# Consensus for the General Use of Equine Water Treadmills for Healthy Horses

**DOI:** 10.3390/ani11020305

**Published:** 2021-01-26

**Authors:** Kathryn Nankervis, Carolyne Tranquille, Persephone McCrae, Jessica York, Morgan Lashley, Matthias Baumann, Melissa King, Erin Sykes, Jessica Lambourn, Kerry-Anne Miskimmin, Donna Allen, Evelyne van Mol, Shelley Brooks, Tonya Willingham, Sam Lacey, Vanessa Hardy, Julie Ellis, Rachel Murray

**Affiliations:** 1Equine Therapy Centre, Hartpury University, Hartpury, Gloucester GL19 3BE, UK; carolyne.tranquille@outlook.com (C.T.); Jessica.Lambourn2@hartpury.ac.uk (J.L.); Kerry-Anne.Miskimmin@hartpury.ac.uk (K.-A.M.); 2Range Animal Science, Office 105, Sul Ross State University, Alpine, TX 79832, USA; persephone.mccrae@sulross.edu; 3Moulton College, West Street, Moulton, Northamptonshire NN3 7RR, UK; jessica.york@moulton.ac.uk (J.Y.); Sam.lacey@moulton.ac.uk (S.L.); 4Sport Horse Health Plan, Jan van Beierenlaan 88, 3445 VV Woerden, The Netherlands; mjjolashley@gmail.com; 5Tränkgasse 4, D-83512 Wasserburg, Germany; info@thissy-baumann.de; 6Department of Clinical Sciences, Colorado State University, 300 West Drake Fort Collins, CO 80523, USA; Melissa.king@colostate.edu; 7Hong Kong Jockey Club, Obe Sports Road, Happy Valley, Hong Kong; erin.v.sykes@hkjc.org.hk; 8Warwickshire College, Moreton Morrell, Warwickshire CV35 9BL, UK; dallen@warwickshire.ac.uk; 9Move to Balance, Terheidedreef 50, 2900 Schoten, Belgium; evelynevanmol@dierfysio.net; 10Langdale Equine, Langdale Farm, Park Lane, Ramsden Heath, Billericay CM11 1NN, UK; shelley@langdaleequine.com; 11Equine Rebalance Therapy Centre, Wellington Riding, Basingstoke Road, Hook, Heckfield, Hampshire RG27 0LJ, UK; equinerebalance@gmail.com; 12Brooksby Melton College, Asfordby Road, Melton Mowbray LE13 7JE, UK; vhardy@smbgroup.ac.uk; 13Department of Nursing, Arden University, Arden House, Middlemarch Park, Coventry CV3 4JF, UK; julie_m_ellis@yahoo.co.uk; 14Rossdales Veterinary Surgeons, Cotton End Road, Exning, Newmarket, Suffolk CB8 7NN, UK; rachel.murray@rossdales.com

**Keywords:** equine, hydrotherapy, water treadmill

## Abstract

**Simple Summary:**

Water treadmill exercise has become popular in recent years for the training and rehabilitation of equine athletes. Water treadmill exercise sessions can be tailored to the individual horse and the training/rehabilitation goals by altering the frequency, duration of exercise, water depth and belt speed. Recent work suggests that there are large variations in current modes of use between users, despite shared training or rehabilitation goals. In 2019, a group of researchers and experienced water treadmill users met in the UK to establish what was commonly considered to be best practice in the use of the modality. The result of these discussions was the production of ‘Water treadmill guidelines—a guide for users’, released in 2020 via various equestrian websites. The purpose of this article is to describe the development of these guidelines and propose them as a starting point for further collaboration between researchers and practitioners in the pursuit of ‘best practice’ in water treadmill exercise for horses.

**Abstract:**

Water treadmill exercise has become popular in recent years for the training and rehabilitation of equine athletes. In 2019, an equine hydrotherapy working group was formed to establish what was commonly considered to be best practice in the use of the modality. This article describes the process by which general guidelines for the application of water treadmill exercise in training and rehabilitation programmes were produced by the working group. The guidelines describe the consensus reached to date on (1) the potential benefits of water treadmill exercise, (2) general good practice in water treadmill exercise, (3) introduction of horses to the exercise, (4) factors influencing selection of belt speed, water depth and duration of exercise, and (5) monitoring movement on the water treadmill. The long-term goal is to reach a consensus on the optimal use of the modality within a training or rehabilitation programme. Collaboration between clinicians, researchers and experienced users is needed to develop research programmes and further guidelines regarding the most appropriate application of the modality for specific veterinary conditions.

## 1. Introduction

Water treadmill exercise is increasingly being used as a routine part of the training of sport horses and racehorses, and more commonly proposed as a component of rehabilitation programmes [1]. Exercise in water offers many potential benefits for equine athletes, including improved aerobic capacity [2], increased joint range of motion through the limbs [3,4] and back [5,6], improved use of arthritic limbs [7,8] whilst minimizing segmental accelerations of the forelimb and decreasing impact shock [9]. These benefits suggest that water treadmill exercise can be a useful addition to a wide variety of training and/or rehabilitation programmes, provided the water treadmill protocol selected elicits physiological and gait adaptations from the horse that align with the particular training or rehabilitation goals. There is still much to be learned regarding optimal use of the modality to best meet the training and/or rehabilitation goals for individual horses. In the meantime, popularity of water treadmill exercise grows, reflected in the number of manufacturers offering equine water treadmills, and considerable improvements in water treadmill design.

A recent survey on equine water treadmill use amongst 41 venues worldwide found that protocols varied widely [1]. The duration of water treadmill sessions varied from 5 to 54 min. Walking speeds reportedly varied from 0.7 to 3.0 m/s, and trot speeds from 3.0 to 5.0 m/s. Arguably, any healthy, sound horse should be capable of walking in fetlock depth water at 0.7 m/s for 20 min (for example) but combinations of high water, and high speed (either in walk or trot) for even a few minutes could present a considerable challenge to an unbalanced, uncoordinated or ill-prepared horse and/or exacerbate injury in a horse with an acute tendon injury (for example). So whilst the survey captured current practice, it does not necessarily reflect best practice. Indeed, fifteen venues within the survey had less than 12 months experience at the time of the survey. The same number (15/40) reported experiencing injury to a horse as a result of the exercise, and 3/40 reported injury to a handler. A rapid increase in the number of new, relatively inexperienced users and lack of formal training programmes in equine hydrotherapy presented a need for guidelines on best practice to ensure horse and handler safety and to protect horse welfare. In 2020, guidelines for best practice in equine water treadmill exercise (see Table 1) were produced by an international equine hydrotherapy working group, comprising experienced users, veterinary specialists and researchers in equine water treadmill exercise. This article describes the process leading to the development of these guidelines.

In 2019, a group of water treadmill users began to discuss the imminent development of accredited qualifications in equine hydrotherapy in the United Kingdom which would incorporate the use of water treadmill exercise. This posed immediate questions about what constitutes best practice in water treadmill exercise and an ‘equine hydrotherapy working group’ was formed with input from invited international colleagues in an attempt to reach consensus on generic guidelines for water treadmill provision for horses. Group membership was established within the first couple of meetings as the founding members (from Moulton College and Hartpury University) invited others to join based on publication history (for researchers) or experience (for practitioners) with an effort made to include international colleagues as far as possible. Many of the group had not met at all prior to these discussions. The aims of the discussions were firstly to provide guidance for new water treadmill users (where ‘user’ refers to those providing water treadmill exercise as a commercial service, or horse owners/trainers seeking to use those services), and secondly, to gain insight on matters where experienced users differed in opinion.

The working group members included researchers, academics, veterinary clinicians and practitioners, and thus combine those involved in generating the evidence relating to water treadmill exercise, those who are responsible for disseminating the research, and those who utilize it. Most of those in the group carried out more than one, if not all of these roles. The members of the group offered a variety of educational and professional backgrounds including qualifications in veterinary medicine, equine physical therapy, veterinary physiotherapy and equine management. The practitioners within the group had experience of using the modality for horses involved in a wide variety of disciplines and levels, including dressage, show jumping, eventing, western disciplines, polo and racing. The group met face to face on several occasions between spring 2019 and 2020, with international colleagues either joining virtually or contributing to the final draft of the guidelines by e-mail. In one of the initial meetings, members shared presentations summarizing either their research findings to date, or their practical experience. Following these presentations of ‘current knowledge’, the group tried to establish the common ground in terms of those aspects of practice that could be agreed upon by all members of the group. Topics discussed were the potential benefits of water treadmill exercise, general good practice in water treadmill exercise, introduction of horses to the exercise, factors influencing selection of speed, water depth and duration of exercise, monitoring movement on the water treadmill and optimal use of the modality within a training or rehabilitation programme.

## 2. Potential Benefits of Water Treadmill Exercise

Equine water treadmill exercise is firmly established as a popular form of cross training for competition horses and a useful form of exercise within equine rehabilitation programmes [10,11,12] due to beneficial effects of immersion, decreased impact shock [9] and gait patterns adopted during water treadmill exercise, namely an increase in range of movement of lower limb joints [3,4] and increased lumbar flexion [5,6]. Currently, only one study has compared walk kinematics of water treadmill and overground exercise in the same group of horses [4]. Immediately following a water treadmill session, carpal and elbow range of motion were shown to be increased, though eight days of water treadmill exercise did not have a lasting effect on overground gait. Similarly, there is not yet any evidence that water treadmill training has any effect on overground locomotion when used regularly within a traditional training programme. However, a number of studies exist describing various chronic physiological adaptation to water treadmill exercise. A recent study found no negative effects of a 112 day water treadmill programme on cartilage metabolism in Quarter horse yearlings transitioning to a 28 day overground training programme, when compared to non-exercise controls or yearlings trained on a dry treadmill for the first 112 days of the programme [13]. To date, there is evidence that 18 days of water treadmill exercise results in a 16% increase in aerobic capacity [2]. After 20 weeks of regular (at least once weekly) water treadmill exercise, subjective assessment of muscle development showed that water treadmill exercise increased development of the gluteal and hind limb musculature more than a control group, suggesting that it may be beneficial for increasing muscle development relating to increased tarsal flexion, but not elevation of the thorax, since cervical trapezius and abdominal development was not significantly different from control [14]. Previous work that utilized a four- and eight-week conditioning protocol did not observe significant cardiocirculatory or muscular property changes [15,16]. Further research on optimal use of water treadmills within training programmes is required. Given that longitudinal training studies are expensive and time consuming in nature, the most valuable studies to industry would be ones that focus on techniques used in practice.

The working group considered that whilst there are potential benefits of water treadmill exercise, incorrect use could either directly oppose the goals of an overland, ridden training programme and/or increase risk of injury. All working group members described the tendency to experiment with the full range of speeds, water depths and, more recently, inclines offered by their particular water treadmill in their early experiences of the modality, but favouring ‘lower’ and ‘slower’ as experience was gained.

## 3. General Good Practice in Water Treadmill Exercise

Water treadmill designs vary, with some machines set into a pit so that the belt is at ground level whilst others are above ground level with a ramped entry and exit; some may only have a rear entry and exit, and there may or may not be a breast/breeching and hind end bar or strap. Given these differences, procedures for loading and unloading were specific to both the particular make/model of treadmill and the environment in which the treadmill is positioned. Loading and unloading procedures, particularly of naïve horses, seemed to present the greatest risk to handlers. A minimum of two handlers was recommended not only for safety, but to keep the horse straight during exercise. Water management is an important consideration, and a water management protocol should be devised based on manufacturers’ recommendations. Specific pool water management training was recommended if chlorinated water is used.

Preparing the horse by brushing the coat, picking out the feet and wrapping up the tail helps keep the water clean. In addition, hosing the horse off prior to entry into the unit will help remove superficial dust/dirt. In doing so, the horse can be checked for any cuts/abrasions that might be below the water level. Upon completion of a water treadmill exercise session, one should consider hosing the horse off again, especially if the water is treated with chlorine/bromine so as to remove chemicals off the hair and skin. Since water can carry microbes leading to infection, it was considered that carrying out water treadmill exercise within four days of a distal limb joint injection was contraindicated. Some studies [1,5] mentioned skin and foot problems as potential consequences of water treadmill exercise. From discussion within the group members, it was apparent that foot problems are more common when the environmental conditions are such that the horse’s feet are never thoroughly dry between sessions, as occurred if the exercise is carried out on a daily/twice daily basis.

## 4. Introduction of Horses to the Exercise

It was agreed that the horses appear to habituate to the exercise more readily in terms of reaching a relaxed and rhythmical gait if they are given several short (up to 15 min) sessions within a short period (three days). The survey by Tranquille et al. (2018) found that 14/41 venues routinely gave horses a light amount of sedation before first exposure to the machine. If used, it was considered by the group that this should be performed following veterinary consultation. Many users kept the water depth low (fetlock level) within the first session, which was the consensus recommendation within the group for initial introduction to the water treadmill.

## 5. Factors Influencing Selection of Speed, Water Depth and Duration of Exercise

As a general guide, speed should be decreased as water depth increases, since drag is increased as water depth increases [17]. Most users exercised horses in walk [1] and walking more slowly than overland was recommended [11,18]. Whilst several of the working group had water treadmills with belt speeds that enabled trot, only one of the 10 venues represented within working group members routinely trotted horses. It was noted that this venue dealt largely with event horses opting to use the water treadmill to enhance fitness. Anecdotally, members of the group felt that benefits of water treadmill exercise could be achieved without trotting. Trotting was believed to cause greater wear and tear on certain types of machine, and necessitate more frequent belt replacement, and this was another reason why, in the absence of positive evidence for the use of trot, it was not favoured by most members.

To ensure safety of horses and handlers, it is advised that horses only trot in the water treadmill once they have become sufficiently habituated to water treadmill exercise at various water heights at a walk. If faster speeds are selected, water depth should be decreased and the gait quality and posture of the horse re-assessed as water depth is incrementally increased. It has previously been shown that increasing water depth has a greater impact on the workload of water treadmill exercise than increasing walking speed [19]. The greatest workload (as measured by oxygen consumption and heart rate) was found to occur at 1.39 m/s in water at the depth of the stifle. However, it is important to note that only a narrow range of speeds (1.11–1.39 m/s) were assessed by this study. This workload (25 mL/(kg·min) is similar to that observed during walking at a speed of 1.6 m/s on a flat, dry treadmill (~20 mL/(kg·min)) [20]. Previous work that relied on heart rate [21] and blood biochemical parameters [22] as an indicator of workload have not shown significant effects of either water depth or speed on workload.

It was recognized that the correct belt speed was horse specific rather than training/rehabilitation goal specific. The choice of water depth was more likely to be training/rehabilitation goal specific. Hence, the best combination of speed and water depth will vary greatly between horses and should be judged according to the individual horse’s response in terms of gait pattern and posture. Hence, it was important to be able to recognize when gait pattern and posture were either not conducive to the training/rehabilitation goals or were indicative of either lameness or muscle fatigue (see “Monitoring movement on the water treadmill,” Table 1).

## 6. Monitoring Movement on the Water Treadmill

The group agreed that assessing posture is essential in determining optimal water depth and speed combinations for each individual horse. Whilst it was recognized that postural assessment is subjective, in the absence of evidence relating posture to muscular adaptation during exercise, it was agreed that the alignment of the head-neck-back and hind limbs during water treadmill exercise should not differ from what would be considered acceptable overground. A large part of the discussion centred around features of the gait pattern during water treadmill exercise that indicated ‘quality’ of movement, and the features that should trigger either a decrease in water depth/speed or an end to the session. Quality of movement was considered to be that which supports and complements a ridden training programme of a sport horse. Straightness and a regular rhythm to the footfalls was deemed important. It was considered that the most common mistake was to combine high walking speed with high water, when many of the benefits of water treadmill exercise can be achieved at relatively slow speed (below that of a comfortable walking speed overland). The belt speed could be considered too high at any given water depth if the horse was seen to have an extended head and neck posture, excessive head and neck displacement (both indicative of increased recruitment of forelimb protractor muscles), extended thoracic posture, increased pelvic axial rotation and increased hind limb retraction (both limiting thoracolumbosacral flexion). Observation of the dynamic posture of the horse during the session was considered to be important to ensure that the horse is ‘pushing’ adequately from the hindquarters rather than relying on ‘pulling’ from the forehand.

## 7. Optimal Use of the Modality within a Training or Rehabilitation Programme

Most working group members considered that water treadmill exercise was useful as a form of cross training for sport horses, and dressage horses in particular, as suggested by the literature [1] when used once or twice a week. It was not possible to gain consensus on the suitability of water treadmill exercise for the rehabilitation of any specific orthopaedic condition without caveats relating to the horse’s conformation, dynamic posture and co-existing lamenesses. As water depth increases, buoyancy increases [23], impact shock reduces [9], and hydrostatic pressure on the limbs increases, all of which have potential benefits for the rehabilitation of certain conditions [7,8,10,11,12]. Evidence pertaining to the use of water treadmill exercise within rehabilitation following treatment for specific injury via controlled studies is limited [7,8] and experimental design challenging, given the highly individual nature of orthopaedic injury, and the tendency for co-existing lameness. Large, multi-centre retrospective case studies might offer the best means of learning more regarding optimal use of water treadmill exercise for rehabilitation. The group suggested that standardization of reporting of protocols with respect to water depth, belt speed, duration and frequency of exercise would increase the opportunity for studies of this nature.

## 8. Conclusions

The guidelines herein state current best practice based on a combination of research evidence base and/or the working group’s combined experience of exercising thousands of horses from a wide range of disciplines. The guidelines were published in the hope that they will be challenged, reviewed and updated as additional evidence emerges. As a result of our discussions, it is apparent that there is much work to be carried out before broad agreement, based on evidence, could be reached regarding the most appropriate application of water treadmill exercise for the rehabilitation and ongoing management of horses with specific orthopaedic injuries. However, by sharing experiences, both positive and negative, practitioners can inform the development of research questions, and research findings can be disseminated directly to users, with benefits for equine welfare and better recovery from injury.

## Figures and Tables

**Table 1 animals-11-00305-t001:** Summary of guidelines for best practice in water treadmill exercise.

General good practice in water treadmill exercise
Follow manufacturer guidelines for correct operation, water care and cleaning of equipment. Seek help from an experienced user to supplement initial manufacturer training. It is recommended all handlers wear personal protective equipment for horse handling. Two handlers are recommended for safety and to keep the horse straight. Devise loading/unloading procedures that avoid handlers being directly in front or behind the horse while on the treadmill belt. Ideally horses should be trained to be led from the side. Before exercise: ○ Examine horse for lacerations, abrasions or skin lesions. When present, water treadmill exercise is contraindicated. ○ Do not perform within 4 days of limb intra-articular injections. ○ If shod: check shoes are secure. Avoid road nails which could damage the belt, and large extensions which may affect foot flight in water. ○ Clean horse to minimise water contamination: brush/wash limbs; pick out and wash feet. Horses can be worked in a headcollar or bridle, provided there is adequate control without restriction of the natural movement of the head and neck. Avoid use of boots on limbs/feet unless there is a specific indication, to avoid risk of frightening horse by boots detaching or moving. Wrap up the tail to decrease water contamination and help horse aftercare. After exercise, wash then dry limbs well. Ensure skin and feet are allowed to fully dry between sessions especially if shod with pads or packing.
Introduction of horses to the exercise
All naïve horses benefit from a structured habituation period; ideally, three sessions of up to 15 min on consecutive days, increasing water depth each day. Ideally the first 2 sessions should be within one week. It is better to avoid starting a habituation process if unable to repeat the exercise within 14 days. Allocate sufficient time for initial sessions to avoid rushing so the horse has a positive experience. Prepare the horse for the belt moving by stepping back and forth; start the belt just as the horse is about to step forward. Be aware that the horse may only notice the introduction of water once it reaches the coronary band. A suitable first session is 10–15 min of relaxed, stable, rhythmical walking in fetlock deep water. If a risk assessment warrants it, light sedation may be useful for the first session under the direction of a veterinary surgeon.
Factors influencing selection of speed, water depth and duration of exercise
The best combination of speed, water depth and duration is affected by individual horse size, stride length, joint ranges of movement and capability, which may change with session duration, recent exercise, fitness or stage of rehabilitation. A suitable belt speed allows the horse to maintain position in the middle of the treadmill leaving room for the head, neck and forelimbs to move without obstruction from the front of the treadmill, breast bar or breast strap. Even small adjustments in speed and/or depth can have significant effects on horse movement. Be prepared to make changes within a session dependent on the horse response. Find a comfortable walk speed for the horse on the treadmill before water is introduced and expect to reduce speed as water depth increases. Be aware that walking in a correct posture through water will be slower than overland movement. Monitoring movement patterns closely is essential. Observe the horse throughout the session, and notice the gait pattern and how the movement changes in response to changes in speed or water depth.
Monitoring movement on the water treadmill
The horse should be straight in line with the treadmill, not positioned/leaning against one side or rolling from side to side. The horse should be able to maintain position in the middle of the belt, neither restricted by the front of the treadmill, or falling to the back of the treadmill. The head and neck should have room to move freely forward and down. The posture should be similar to overland walking, with the face in front of the vertical, a ventroflexed (rounded) lumbar spine and the hind limbs stepping well under the body. An extended (dorsiflexed) position of the cervical, thoracic and lumbar spine, with the face approaching horizontal and hind limbs trailing should be avoided. The head should be largely still, without excessive horizontal or vertical displacements or head tossing. The hind limbs should be placed in the path of the forelimbs, and not increasingly medially or laterally as the water depth is increased. There should be a regular rhythm to the footfalls. If movement quality is not achieved or reduces during a session, then speed and water depth should be re-evaluated, and/or duration of the session modified.
Optimal use of the modality within a training or rehabilitation programme
Water treadmill exercise provides straight line, unridden, controlled exercise that provides an option for cross training alongside a normal training programme. It is not recommended as the only or primary exercise type unless specifically indicated for rehabilitation of a particular injury, generally for a limited time period. Drag increases as water depth and stride frequency increase, which has the potential to limit limb protraction, alter muscle use and change stride pattern. As water depth increases, impact shock of the limbs is reduced. Water treadmill exercise in walk or trot does not produce high heart rates, so fatigue may not be obvious but horses may still experience fatigue in certain muscle groups, which is important to consider when planning a programme or monitoring within a session. Oxygen consumption has been found to reach 20% of maximal oxygen consumption in stifle depth water. Consider a water treadmill exercise session as equivalent to a challenging ground schooling session.

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
