# Peer review of "Consensus for the General Use of Equine Water Treadmills for Healthy Horses"

_animals, 2021, doi:10.3390/ani11020305_

Round 1

Reviewer 1 Report

As indicated by the authors, this manuscript describes the conclusions obtained by a multifactorial working group with experience in the use of WT in various fields of equine training and rehabilitation. The article has been structured in the following parts: 1) Benefits of use; 2) Good practices in its management; 3) Introduction to exercise in WT; 4) Factors that influence the speed, depth and duration of exercise in WT; 5) Movement monitoring; 6) Optimal use of a modality in a rehabilitation or training program.

Main comments

I understand that this article has the objective of providing quick information to researchers / clinicians / trainers who will be using the WT in the future for various training or rehabilitation purposes. Therefore, I believe that if the journal allows it, its publication could be considered a technical note, or a presentation of results derived from a workshop.

First, if the aims of the authors is to provide guidelines for uses of WT directed to people involved in the equestrian world, as trainers, the manuscript appears to be useful, but in my opinion, it should be published in other types of journals, with greater social relevance. In this case, I believe that this article would include general aspects such as good management practices, the introduction to exercise or the factors that determine the speed, duration of exercise and water depth. The other aspects should be developed for people specialized in equine sport medicine and rehabilitation. In addition, I think it is essential to increase the information you provide in some points, as I will indicate later in the other comments section.

Second, if the authors want to publish the MS in a scientific journal, particularly for researchers, equine clinicians or rehabilitating veterinarians, I think it should be of higher scientific quality, presenting the research results that we currently have. In this case, concrete research results should be showed on the various aspects they address. For example, in the movement monitoring section, they do not indicate what results have been found or how they can be monitored. Likewise, the last point, regarding the optimal use of this type of exercise, does not provide any specific information.

Specific comments

Title. The title does not adequately reflect the content. The manuscript seems to be aimed to describe the correct procedures for the physical use of the WT, in order to guarantee the welfare and safety of the horses and to avoid trauma to handlers and horses. It does not refer to possible rules for its use in specific rehabilitation or training programs, depending on the proposed objectives.

Lines 68-69. According to the authors, who make reference to its previous article, the protocols of use are highly variable, presenting maximum and minimum values (for example, duration of the exercise session in WT from 5 to 54 minutes). The authors do not explain the cause of this. Obviously the duration of the exercise, as well as the other factors that the authors consider in these lines, will depend on the type of horse and the objective of using the WT. The duration for the rehabilitation of a tendonitis, particularly in a horse that has been out of training will not be the same as a training session. To avoid confusion to the reader who has no experience on the subject, it is essential to clarify this aspect.

Lines 111-112. While WT training is well established as a popular form of cross-training, the authors should clarify that its effects on performance enhancement when included in a regular training program have not yet been evaluated, and it should be investigated in the future. There is some isolated research on training on a WT (Borgia et al., 2010; Firshman et al., 2015; Silvers et al., 2020). It would have been interesting if the authors add in some information derived from these articles.

Lines 112-113. The authors should clarify that many of these findings have been found exclusively during one or more exercise sessions in the WT, but their effect on terrestrial locomotion has not been subsequently proven. For example, in articles 3 and 4 only the kinematics on the WT have been studied. Of course, this does not mean that the WT is invalid, but that its usefulness should be demonstrated. The reader must be clear about this.

Lines 148 to 160. The authors do not present specific data on the factors that influence the selection of speed, water depth or duration of exercise. Many of the statements are based on personal experiences, which can be highly subjective.

Lines 161-on. The same as in the previous case. It is not specified what is quality of movement, or how it is determined, or for what type of horses. The information provided is very unspecific and highly subjective. For example, considering lines 172-174: the authors indicate that observation of the horse's posture is important. Certainly, but it should be evaluated in a more scientific way than a simple postural observation.

Lines 175-on. The authors do not provide concise information on the optimal use of WT within a training or rehabilitation program.

Author Response

Thank you all for your time and effort in providing valuable feedback on our manuscript. We have considered all your comments and suggestions; and we have made the following changes as a result.

Reviewer 1:

In answer to the Main Comments:

On consideration of your comments, we have added more detail to the sections on ‘Factors influencing selection of belt speed, water depth and duration of exercise’; ‘monitoring movement on the water treadmill’ and ‘optimal use of the modality within a training and rehabilitation programme’.

Specific Comments:

We have altered the title to more accurately reflect the content “Consensus for the General Use of Equine Water Treadmills for Healthy Horses”.

Lines 68-69: Whilst we cannot explain why some people select the exercise protocols they do, we have hopefully addressed the point you make with additional text lines 75-80 in resubmitted version.

Lines 111-112 and 112-113: Indeed, the effects of WT training on performance when incorporated into a training programme have not yet been established. We have now made this clear (lines 134-136). This section has been expanded (lines 131-151) presenting studies which have described effects of water treadmill training programmes on cartilage metabolism, aerobic capacity, muscle development and muscle fibre characteristics (new refs 13, 14, 15 and 16).

Lines 148-160: This section has been expanded and reorganised with more detail added (20, 21, and 22 added). Discussion around decisions on depth/speed revealed how important working group members felt it was to judge each horse on an individual basis. The reason for the section on monitoring movement came as a direct result of this discussion. We hope this is clearer now in the revised section lines 193-224 in resubmitted version.

Line 161-on: Monitoring movement on the water treadmill: We have added more detail lines 227-231 and 240-242 explaining why ‘posture’ was felt to be important even though it is subjective.

Lines 172-175: We have acknowledged that postural assessment is subjective and visual (lines 235-238) but it was considered to be useful.

Lines 175-on: We have attempted to give more clarity in this section, stating simply that there is, as yet, no agreement on how best to use the modality for specific musculoskeletal conditions.

Reviewer 2 Report

  • A brief summary 

This paper provides a summary of the making of the guidelines for best practice in water treadmill elaborated by a group of researchers during a meeting in 2019 and also the guidelines elaborated by them in a table. These guidelines have already been published somewhere else. They conclude that these guidelines present the current best practice base on research evidence and the wide experience on the use of the water treadmill of researchers presenting this paper.

  • Broad comments 

This paper presents a very useful guidelines for user of water treadmill and researchers show their concern about the need of more research in this area. In my opinion the purpose of this article to describe the background to the production of this guidelines should be addressed in a more efficient way, please look at the specific comments. On the other hand, it does open a further collaboration between researchers and practice although it will be interesting for the scientific world if they make a list of recommendation of future research based on the needs observed by this group.

  • Specific comments 

In table 1: in the first point when it says: “Examine horse for lacerations, abrasions or skin lesions, when water treadmill exercise may be contraindicated” I believe that they should be more specific, it is contraindicated or not. In my opinion if the water is reused any horse with an open wound must not used the water treadmill because it may get an infection.

In lines 115 to 118, the authors are concerns about the users of water treadmill not achieving maximal benefit of the horses in their care. How do the measure that? If it is just an impression, here they should recommend future research in this area for example.

In lines 138 and 139, it will be interesting if they also present data about this affirmation.

In line 158, the researchers present that only 1 of 10 venues trotted their horses. In our venue we also don´t trot our horses because we are afraid the horses may slip. Was there any discussion about this topic? It seems they express that horses may trot, but what guidelines should water treadmill follow before trotting their horse on the water treadmill?

Author Response

Thank you for your time and effort in providing valuable feedback on our manuscript. We have considered all your comments and suggestions; and we have made the following changes as a result.

Table 1: thank you – this has been altered (Table 1) to be more directive “Examine horse for lacerations, abrasions or skin lesions. When present, water treadmill exercise is contraindicated”

Lines 115-118: We have elaborated on this in lines 75-80 in the revised version and lines 153-158 which we hope makes it clearer that inappropriate use of WT exercise could cause injury.

Lines 138-9: We do not have data, but discussion revealed foot/skin problems only occurred with frequent (daily) use, so again we have provided this detail (line 180-182)

Line 158: We have addressed this in lines 199-203. Incidentally re fear of horses slipping in your own venue – it is worth getting your manufacturer to check the belt tension. In one of the authors’ experience (KN) horses don’t slip, but belts can! Habituation to trot was not discussed specifically, since there was only one venue that used it.

Reviewer 3 Report

Thank you for putting together this piece of work. This is a wonderful start to guidelines for this important form of equine exercise.

I have included a few suggestions on the pdf itself. In general, I believe that this manuscript will benefit from expansion on the discussions of each topic. In the current state there is very little information presented beyond the actual guidelines. To provide more background it would be useful to the reader to see more of what was discussed and how indeed the guidelines were developed - what were topics that people had differing opinions on and how were they handled? Why were some aspects included but not others (example water height and speed - guidelines are not very specific however, one can imagine that for specific breeds / heights more specific guidelines could be provided for belt speed and water height depending on goal of exercise (CV training, muscle strength development).

Author Response

Thank you for your time and effort in providing valuable feedback on our manuscript.

Thank you also for the helpful comments and suggested edits in the pdf, these have been addressed in track changes within the revised doc. We hope that the additional detail provided in response to reviewer 1 and 2’s comments also provide more specific detail as to the nature of the discussions (as you requested). We have been more explicit particularly with regard to the use for rehab where no consensus could be reached. Specific guidelines could not be given for the most appropriate water depth for any given purpose because this was deemed to be ‘horse-specific’, hence the section on ‘monitoring movement’. We hope the revisions make this clear.

Round 2

Reviewer 2 Report

Thank you very much for the improvement of your paper. The suggestions of the other reviewers make this paper even more useful and interesting. About your recommendation of getting our manufacturer to check the belt tension, our horses are not slipping at walk, but we do not know at trot because we do not use the WT in this gait but thank you for your recommendation.

Line 136 or 137 should be eliminated because they are repeated.

These are only suggestions that you may use if you like.

Lines 157 to 158 I would not say “the most valuable studies to industry would be the ones that focus on techniques used in practice” but -some studies that may be also useful are the ones that focus on the benefices of different techniques on the performance or rehabilitation of the horses-

Line 165. You may add, but research with sufficient relevant data is still needed to verify our experience.

Line 190, It will be nice if you publish this data, if we can see that in numbers.

Lines 214 to 225. In this paragraph you may also support your idea with this paper: Saitua, A., Becero, M., Argüelles, D., Castejón-Riber, C., Sánchez de Medina, A., Satué, K., & Muñoz, A. (2020). Combined Effects of Water Depth and Velocity on the Accelerometric Parameters Measured in Horses Exercised on a Water Treadmill. Animals10(2), 236.

Author Response

Thank you so much for taking the time to read the resubmission. We're pleased you approve of the changes. Thanks also for the further suggestions, which we have considered, and our responses are below.

Line 136 or 137 should be eliminated because they are repeated.

Response: Thank you. This is done

These are only suggestions that you may use if you like.

Lines 157 to 158 I would not say “the most valuable studies to industry would be the ones that focus on techniques used in practice” but -some studies that may be also useful are the ones that focus on the benefices of different techniques on the performance or rehabilitation of the horses-

Response: Thank you, but we do actually wish to say that we feel it is preferable to study those protocols already used commonly by industry.

Line 165. You may add, but research with sufficient relevant data is still needed to verify our experience.

Response: Thanks again, this statement is an accurate reflection of the discussion, which we hope is valuable in itself, provided the reader understands that it reflects the experiences of this working group, hence why we made sure to start the sentence 'All working group members....'. 

Line 190, It will be nice if you publish this data, if we can see that in numbers.

Response: There was no 'data' as such to support this, but we'll bear it in mind for future.

Lines 214 to 225. In this paragraph you may also support your idea with this paper: Saitua, A., Becero, M., Argüelles, D., Castejón-Riber, C., Sánchez de Medina, A., Satué, K., & Muñoz, A. (2020). Combined Effects of Water Depth and Velocity on the Accelerometric Parameters Measured in Horses Exercised on a Water Treadmill. Animals10(2), 236.

Response: Thank you. We were aware of this paper, but the section is on trot (?), and this study we presume was walk, hence why some horses could not maintain position on the belt at top speed and depth.

Reviewer 3 Report

This is much improved and i like the new title. There are only a few spelling and formatting problems that will need to be addressed prior to publication.

Author Response

Thank you for taking the time to re-read our paper. We're very grateful for your input!